# An integrated augmented reality surgical navigation platform using multi-modality imaging for guidance

Harley H. L. Chan[1]*, Stephan K. Haerle[2], Michael J. Daly[1], Jinzi Zheng[1], Lauren Philp[3,4], Marco Ferrari[1,5,6], Catriona M. Douglas[1,5,7], Jonathan C. Irish[1,5,7]

1 TECHNA Institute, University Health Network, Toronto, ON, Canada, 2 Center for Head and Neck Surgical Oncology and Reconstructive Surgery, Hirslanden Clinic, Lucerne, Switzerland, 3 Institute of Medical Science, University of Toronto, Toronto, ON, Canada, 4 Department of Obstetrics and Gynecology, University of Toronto, Toronto, ON, Canada, 5 Department of Otolaryngology-Head and Neck Surgery, University of Toronto, Toronto, ON, Canada, 6 Unit of Otorhinolaryngology–Head and Neck Surgery, University of Brescia, Brescia, Italy, 7 Department of Surgical Oncology, Princess Margaret Cancer Centre, University Health Network, Toronto, ON, Canada

* Harley.Chan@rmp.uhn.ca

**Data Availability Statement:** All relevant data are within the manuscript and its Supporting Information files.

## Abstract

An integrated augmented reality (AR) surgical navigation system that potentially improves intra-operative visualization of concealed anatomical structures. Integration of real-time tracking technology with a laser pico-projector allows the surgical surface to be augmented by projecting virtual images of lesions and critical structures created by multimodality imaging. We aim to quantitatively and qualitatively evaluate the performance of a prototype interactive AR surgical navigation system through a series of pre-clinical studies. Four pre-clinical animal studies using xenograft mouse models were conducted to investigate system performance. A combination of CT, PET, SPECT, and MRI images were used to augment the mouse body during image-guided procedures to assess feasibility. A phantom with machined features was employed to quantitatively estimate the system accuracy. All the image-guided procedures were successfully performed. The tracked pico-projector correctly and reliably depicted virtual images on the animal body, highlighting the location of tumour and anatomical structures. The phantom study demonstrates the system was accurate to 0.55 ± 0.33mm. This paper presents a prototype real-time tracking AR surgical navigation system that improves visualization of underlying critical structures by overlaying virtual images onto the surgical site. This proof-of-concept pre-clinical study demonstrated both the clinical applicability and high precision of the system which was noted to be accurate to <1mm.

## Introduction

Imaging-based surgical navigation (SN) systems are routinely used to guide surgical procedures in anatomically complex areas such as the head and neck [1]. Previous studies have demonstrated that the use of SN can improve efficiency and safety in these challenging areas [2].

**Funding:** This work is funded by the Guided Therapeutics (GTx) Program-TECHNA Institute, University Health Network, Kevin and Sandra Sullivan Chair in Surgical Oncology, Hatch Engineering Fellowship Fund, and Princess Margaret Hospital Foundation. IDEAS grant by Radiation Medicine Program, Princess Margaret Cancer Centre.

**Competing interests:** The authors have declared that no competing interests exist.

The added value of SN is twofold: firstly, it facilitates the identification of critical anatomical structures to avoid unnecessary complications; and furthermore, it helps to delineate tumor boundaries during oncologic ablations with the intent to improve adequacy of margins [3–5].

Currently, research is focused on augmented reality (AR) methods such as video-computed tomography (CT) augmentation [6, 7] and intraoperative imaging [8] to further improve the usefulness of SN. Optical "see-through" techniques, such as video-CT, consist of generating virtual anatomical structures from cross-sectional images (e.g. CT or magnetic resonance imaging [MRI]) that are overlaid on the endoscopic image [9]. Literatures reveal that augmented reality in medical research had been very active in past decade [10]. However, majority of proposed approaches are AR system independently operate and not integrate into surgical navigation system. Nowadays, the surgical navigation system has been routinely used in the interventional procedure. The integration of SN with AR would synergistically improve the performance of technology based guidance. In fact, an integrated AR-SN system would provide a precise, real-time topographical localization of the surgical field by means of intuitive visual enhancement.

Following this line of research, the Guided Therapeutics (GTx) group (TECHNA Institute, University Health Network, Toronto, Ontario, Canada) has developed a handheld AR device integrated into a SN system, which is capable of surgical site augmentation using both medical images and computer-generated virtual images. Briefly, this augmentation is achieved by using a tracked pico-projector which superimposes pre-contoured structures (i.e. the volume occupied by the tumour, or critical neural and vascular structures) onto the surgical field. The aim of this was to assess the feasibility of this novel imaging system (AR-SN integrated) through pre-clinical mice models.

## Materials and methods

### System architecture

The prototype AR-SN system consists of pico-projector (L1 v2, AAXA Technology Inc., Santa Ana, California, United States), an infrared (IR) real-time optical tracking system (Polaris Spectra, NDI, Waterloo, Ontario, Canada), a universal serial bus (USB) 2.0M pixels generic camera, and a laptop computer. The pico-projector employed in the prototype is light (170g) and physically small (54mm x 170mm x 21mm).

The enclosure of the AR device is fabricated with acrylic and designed to encase the pico-projector, the IR reference marker mount, and the USB camera. The signal transmission of pico-projector is via High-Definition Multimedia Interface (HDMI) connector. Data processing is performed using a laptop computer (M4500, Precision laptop, Dell, Round Rock, Texas). Fig 1 illustrates the spatial relationship and interaction between individual components in the augmented reality surgical navigation platform.

A SN platform, named "GTxEyes" (in-house development) was used in combination with the prototype as it allows for image display, fusion, and overlay of multiple imaging modalities including standard CT, cone beam CT, MRI, single photon emission CT (SPECT), positron emission tomography (PET), and video-endoscopy. The GTxEyes platform [11] was developed using open-source, cross-platform libraries included IGSTK [12], ITK [13], and VTK [14].

Surgical plans including the target lesion, ablation margins, critical structures, and safety margin (i.e. voxel contouring of structures to be spared during ablation) were created pre-operatively using ITK-SNAP [15] and built on the SN system. Real-time tracking detects the location of the surgical instruments in 3D space, thus guiding the surgeon throughout the ablation and can alert the surgeon when the navigated instrumentation enters a pre-determined safety margin volume [16]. The AR-SN platform supports fully automatic bony and soft tissue

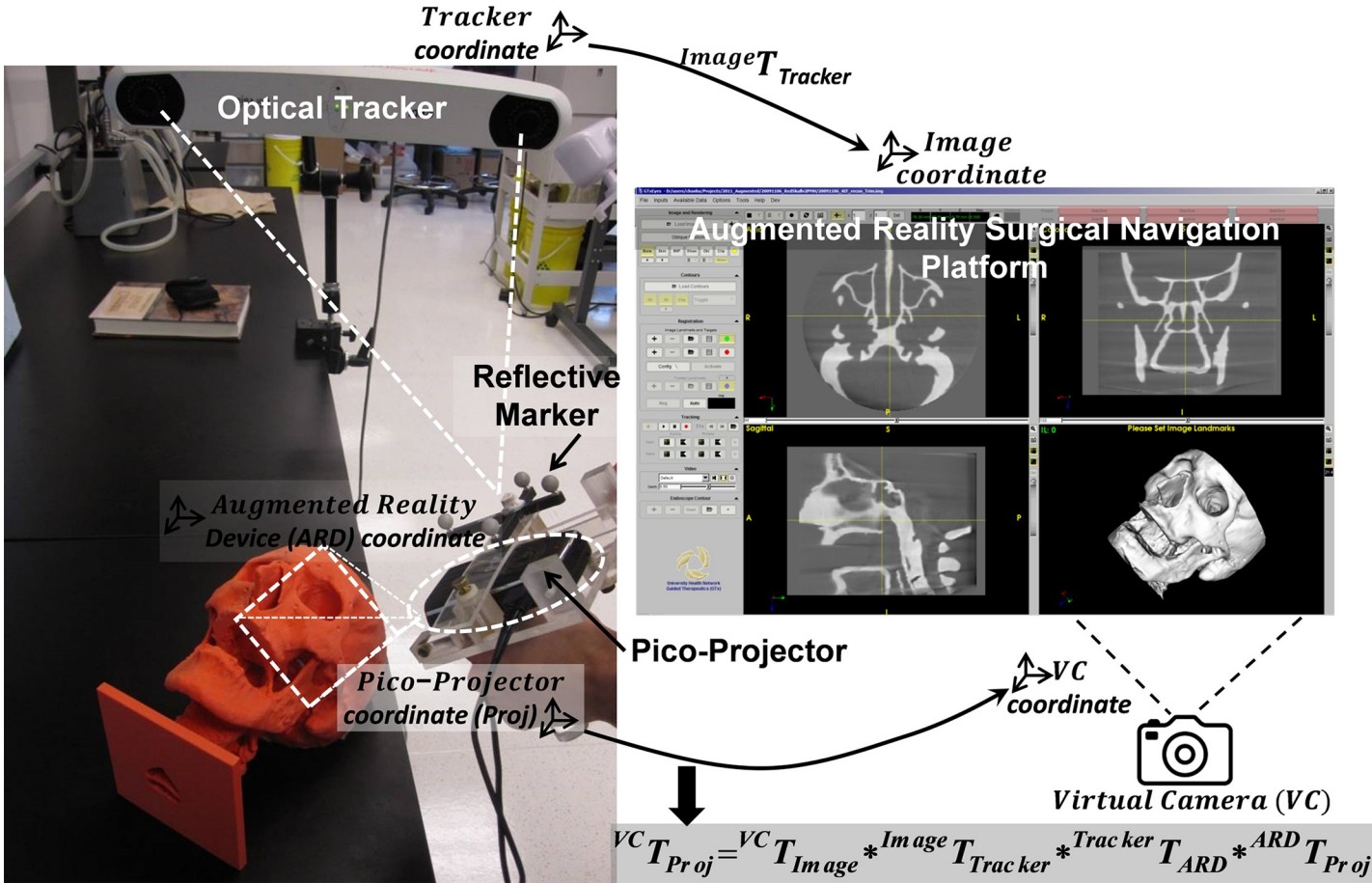

**Fig 1. Prototype augment reality surgical navigation platform consists of optical tracking system and tracked pico-projector.**

digital segmentation based on voxel intensity threshold value (e.g. Hounsfield Unit for CT imaging) and all surgical plans can represented by either 3D surface or volume rendering and overlaid onto the surgical field with adjustable opacity. The AR-SN platform allows the surgeon to scroll through image slices projected on the operative field with a tracked pointer and to accordingly decide the depth of images through which to augment the surgical view.

## Image overlay and system operation

The AR-SN system was registered into a single coordinate system by pairing correspondent landmarks using fiducial markers identifiable in both the image and subject. Once registration was completed, the AR-SN platform could track multiple surgical instruments simultaneously, as described in a previous study [17, 18].

To facilitate real-time tracking of the AR device in 3D space, an optical sensor attachment was mounted to the enclosure of the pico-projector. Pre-operative calibration consecutively performs to define the spatial relationship between the sensor and the centre of the pico-projector, which elaborates by a transformation matrix stored in the AR-SN platform. Additionally, calibration allows tracking of the spatial position of the projector and synchronization of a virtual camera in the AR-SN platform. This calibration procedure ensures the pico-projector correctly illuminates the surgical field providing a reliable image of the pre-contoured structures (i.e. those delineated in the surgical plans). Further details on the calibration and

registration procedure are provided in the S1 File. By virtue of the real-time tracking, the pico-projector can be repositioned during the surgical procedure according to intraoperative requirements without compromising projection accuracy. Moreover, the integrated AR-SN can project multiple virtual objects and render multimodality fused imaging. Contours from external software such as ITK-SNAP [15] and 3D slicer [18, 19] can also be imported into the AR-SN platform for sophisticated delineation of anatomical structures.

## Preclinical animal studies

The integrated AR-SN system is evaluated on four preclinical mice models. All animal studies are performed in the animal care resource facility at the Princess Margaret Cancer Research Centre in accordance with protocols approved by the animal care committee at University Health Network. Four independent studies are performed to evaluate system performance and to investigate multimodality imaging in intraoperative AR-SN-based guidance. The first two studies investigated PET/CT for AR image guidance, whereas the third and fourth studies investigated PET/MRI and SPECT/CT, respectively.

In the first study, $^{64}$Cu isotope (10–13 MBq per mouse) was administered to a healthy, immunocompetent CD-1 mouse (Charles River, Wilmington, Massachusetts, United States) intravenously via a lateral tail vein one hour before micro-PET imaging (Focus 220, Siemens, Malvern, Pennsylvania, United States). A whole body image acquisition time took 15 minutes per mouse, the image resolution was 0.146 x 0.146 x 0.796 mm$^3$. CT images were acquired after PET-imaging using a micro-CT scanner (Locus Ultra, GE Healthcare, Pittsburgh, Pennsylvania, United States) with imaging parameters set at 80 kVp and 50 mA. During CT scanning, multiple 3D printed polycarbonate fiducial markers were introduced to surround the mouse bed to facilitate the subsequent AR-SN system registration. The resulting image volume was 366 x 196 x 680 voxels (366 x 196 matrix with 680 slice images), with isotropic voxel size of 0.154 mm$^3$. PET/CT images were then co-registered using Inveon Research Workplace (IRW) software (Siemens Healthcare, Malvern, Pennsylvania, United States). CT images were downsampled to 0.3 mm$^3$ isotopic voxel size to minimize computational intensity.

In the second study, 5 million BT474-M3 cells (HER2-overexpressing breast carcinoma) suspended in 100μL PBS were injected subcutaneously into the right mammary fat pad of athymic female CD-1 nude mouse. The animal was then monitored bi-weekly for tumour growth. Once the tumor reached a size of ~400-500mm$^3$, the animal underwent the PET/CT as previously described. The tumour was preoperatively contoured based on available imaging using ITK-SNAP [15], and subsequently imported into the AR-SN platform.

Third study was investigating PET/MRI AR image guidance, a female CD-1 nude mouse was inoculated with 5 million MDA-MB-231 breast cancer cells through a subcutaneous injection in the bilateral upper mammary fat pads (100μL on each side). The mouse was then monitored as previously described. Once tumours reached a size of approximately 200-250mm$^3$, liposomes loaded with $^{64}$Cu were administrated intravenously 24 hours prior to imaging at a dose of 10–13 MBq/mouse and 20 μmol phospholipid/kg [20]. The mouse then underwent micro-PET imaging with a longer acquisition time of 60 minutes. Whole thorax-abdomen MRI was then performed using a 7T micro-MRI scanner (M2, Aspect Imaging, Shoham, Israel). The fusion of the MR and PET data sets was registered using IRW software with a rigid body algorithm based on normalized mutual information.

The fourth study experimented SPECT/CT image modality for AR image guidance. A female athymic CD-1 nude mouse were injected 10 million 231-H2N cells (human breast cancer cells) into the left thigh. The mouse was monitored as previously described until the tumour reached a volume of 125 mm$^3$. The mouse was then injected intravenously with 37

MBq of [111]In-Fab fragments of anti-HEGF (human epidermal growth factor) antibody via lateral tail vein injection 48 hours prior to imaging [21]. SPECT/CT imaging (Bioscan, Washington DC, Washington, United States), was performed with dual-modality machine. Photons were accepted from the 10% windows centered on indium two photo-peaks at 171 and 245 keV. The SPECT projections were acquired in a 256 x 256 x 16 matrix for 85 minutes. Voxel size was isotropic 0.3mm$^3$. Images were reconstructed using an ordered-subset expectation maximization algorithm (9 iterations) [22]. Cone-beam CT images were acquired (180 projections, 45 kVp) immediately before the micro-SPECT images. Eight 3D printed markers were attached around the scan bed for tracking registration. Co-registration of SPECT and CT images was performed using pertinent software (InvivoScope, Bioscan Inc, Washington DC, Washington, United States).

Upon completion of all multimodality imaging studies, mice were euthanized using an overdose of 5% inhaled isoflurane and their body position was rigidly maintained in the scan bed. For each experiment, the AR device was mounted on an articulating arm located at approximately 250–350 mm above the supine mouse.

## AR-SN system accuracy measurement

The accuracy of the AR-SN system was quantitatively evaluated by overlaying the projection image on a checkerboard phantom with known dimensions. The phantom was composed of 25 rectangular grids (10 x 10 mm$^2$ each) with central divots (5 rows and 5 columns). The distance between the central divots was 20mm. The acrylic phantom was fabricated by a high precision computer numerical control (CNC) machine. The guidance images for the phantom navigation were acquired with a prototype cone-beam CT C-arm [23]. CT images were 256 x 256 x 192 voxels by volume with an isotopic voxel size of 0.78 mm$^3$. The accuracy of the AR-SN system was evaluated by comparing landmark localization on the phantom with and without the use of AR guidance. The central divot of each grid was localized manually using the tracked pointer three separate times per grid. The represented location of the divot was calculated from the mean location of the tracked pointer over the three measurements. These localization exercises were conducted over a range of distances between the phantom and AR projector including 200, 300, and 400 mm. The uncertainty of the projected location was calculated as follows: $(x_i, y_i, z_i)_{real}$ represents as "real" location of the divot center acquired manually without AR image guidance and $(x_i, y_i, z_i)_{virtual}$ represents as "virtual" location of the divot generated from the AR projection image, the estimation of error is the distance between the "real" and "virtual" location of the central divot such that $Err = \|(x_i y_i z_i)_{real} - (x_i y_i z_i)_{virtual}\|$.

## Results

### Preclinical animal study

**First and second study: AR-SN with PET/CT.** In both the first and second studies, the AR-SN system provided reliable augmentation of 3D virtual skeleton (in gray) and a semi-opaque coronal slice of PET image. Fig 2 demonstrates the experiment setup and AR projection images indicate skeleton and uptake of isotope from PET image in the liver and bladder.

In the second study, the right mammary fat pad tumour was delineated with ITK-SNAP [15] and imported to the AR-SN system in addition to the PET/CT images. The content of AR projection images (Fig 3A) included fusion with semi-opaque coronal slices of both CT (in gray) and PET (in hot colour scale) images. Additionally, a surface rendering of 3D bone structures (in gray) was also projected on the surgical field. The 3D virtual tumour on the right mammary fat pad (Fig 3A, in green) distinctively highlighted the tumour location relative to the other anatomy on PET/CT slice. Fig 3B demonstrates the virtual image created through

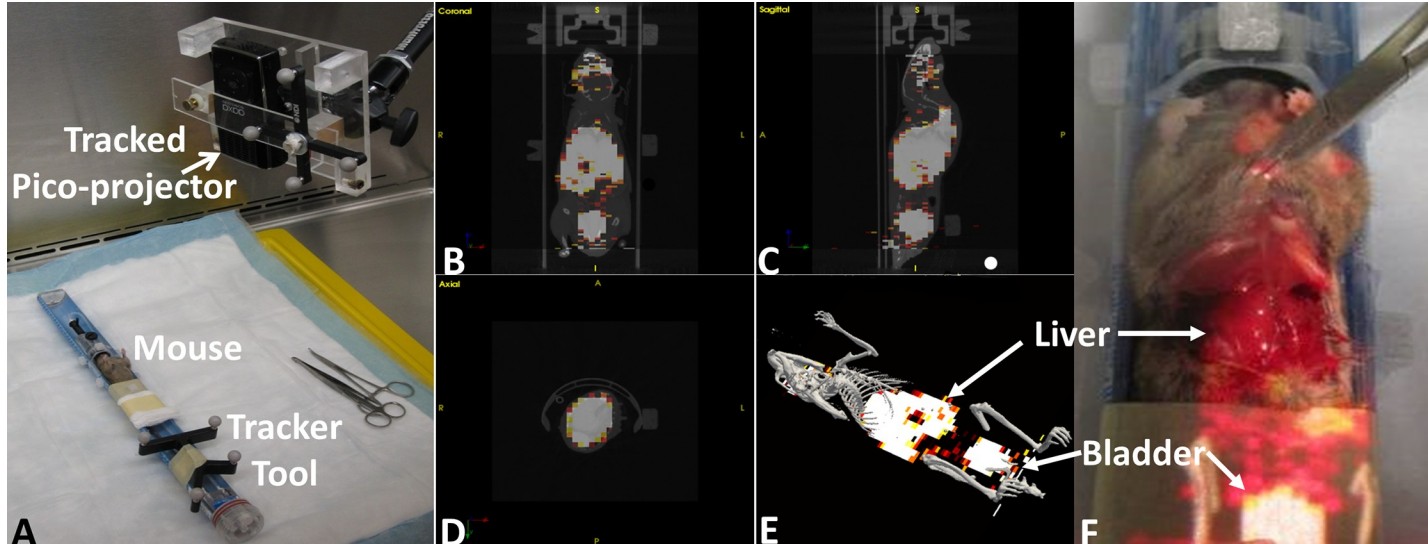

**Fig 2. Augmented reality guided procedure with immunocompetent CD-1 healthy mouse model using PET/CT image.** (A) experimental setup (B), (C) and (D) tri-planer view of coronal, sagittal and axial imaging slices respectively. d) virtual image containing 3D skeleton (gray) and coronal slice of PET image. (F) showing the image projection on the abdominal surface of dissected mouse indicate the accumulated isotope in the liver and bladder.

projection of a PET/CT coronal slice onto the mouse body, with the virtual tumour seen in green and the mouse skeleton seen in gray. The surgical dissection of the mouse (Fig 3C) demonstrated that the anatomical tumor location matched the location identified by the augmented image overlay on the mouse body (Fig 3D). This confirmed the correlation of the virtual tumor projection and the anatomical findings at the time of surgery and demonstrated the precision of the AR instrument in physical space.

**Third study: AR-SN with PET/MRI.** The registered PET/MRI was imported into the AR-SN system. The content of the augmented reality projection image included a fused PET/MRI coronal slice (Fig 4). The projected PET image displayed increased signal intensity signifying $^{64}$Cu-isotope uptake by the bilateral fat-pad tumour which was precisely overlaid onto the anatomical location of the tumours. The projection image also highlighted the high-level lung uptake of the $^{64}$Cu-radioisotope (Fig 4).

**Fourth study: AR-SN with SPECT/CT.** SPECT/CT DICOM data was imported to the AR-SN navigation platform and fiducial markers were used to register the images with the AR platform. The content of the AR projection image included a semi-opaque fused SPECT (in rainbow) and CT (in gray) Coronal slice (Fig 5B). The axial slice and sagittal slice of the same mouse is showing in Fig 5A and 5C, respectively. In addition, the surface rendering of the full mouse skeleton (Fig 5D) created using bone segmentation was virtually projected in purple. Fig 5E also highlights the high-level isotope uptake in the liver.

### AR-SN system accuracy measurement

The accuracy measurement proceeds with determining the divots 3D location with and without AR guidance in a range of projection distance from 200mm to 400mm. The fiducial registration error (FRE) with 4 divots is 0.45mm. After tracker-image registration, a tracked pointer is used to localize the central of divot per described in the method section. Fiducial localization error (FLE) represent by root-mean-square (RMS) reveal the discrepancy between real divot and virtual divot location. At 200mm, 300mm and 400mm projection distance, the RMS of measured divots is 0.42 ± 0.25 mm, 0.53 ± 0.31mm and 0.7 ± 0.37 mm, respectively.

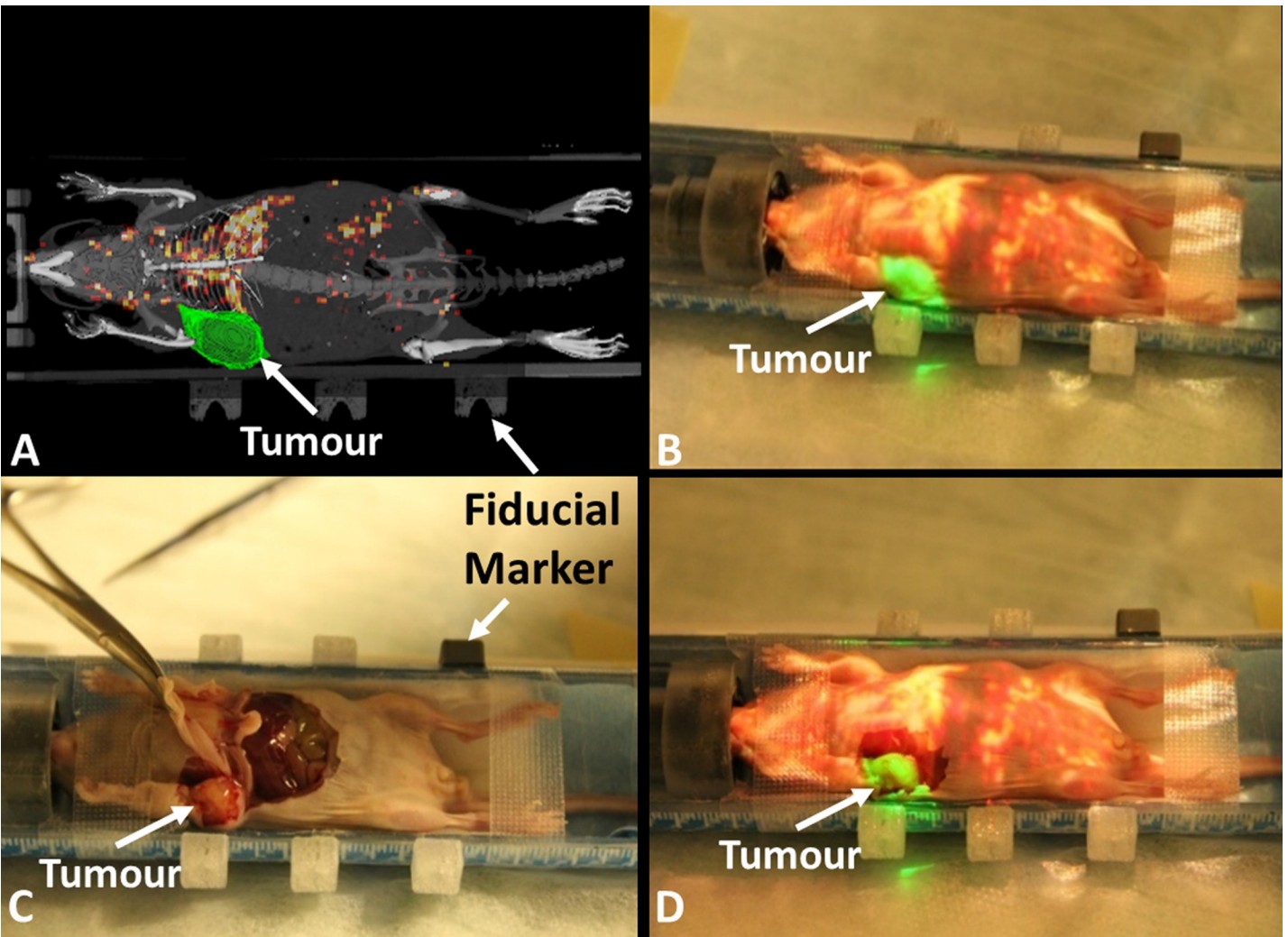

**Fig 3. Augmented reality guided procedure with the breast carcinoma xenograft model.** (A) projection image includes semi-opaque fused micro-CT and PET images with contour of tumor in green and skeleton of mouse in gray. (B) projection image overlaid on top of the mouse indicates the tumor location (green) and highlights isotope uptake in the liver. (C) mouse dissection demonstrating the tumor location. (D) mouse dissection with the overlaid projection image demonstrating the correct anatomical localization of the virtual contoured tumor on the mammary fat pad tumour.

The overall system RMS across various projection distances was 0.55 ± 0.33 mm. Fig 6A illustrates the checkerboard phantom partially augmented by the projection image of 3D virtual checkerboard generated from CBCT. The projection demonstrated remarkable spatial accuracy and correlation between the physical location of the phantom central divots and the projected AR image (Fig 6B). The FLE for each projection distance is summarized in Fig 6C. The AR system performed best at projection distance of 200mm and progressively worsened increasing the distance to 300 and 400 mm (Fig 6C).

## Discussion

This study demonstrates the feasibility and accuracy of our novel AR-SN system prototype. This system is composed of AR device fully integrated with a real-time SN system and is capable of providing target anatomical localization with an accuracy of 0.55 mm. Our implementation of AR technology distinct from existing approaches that are published in the literatures

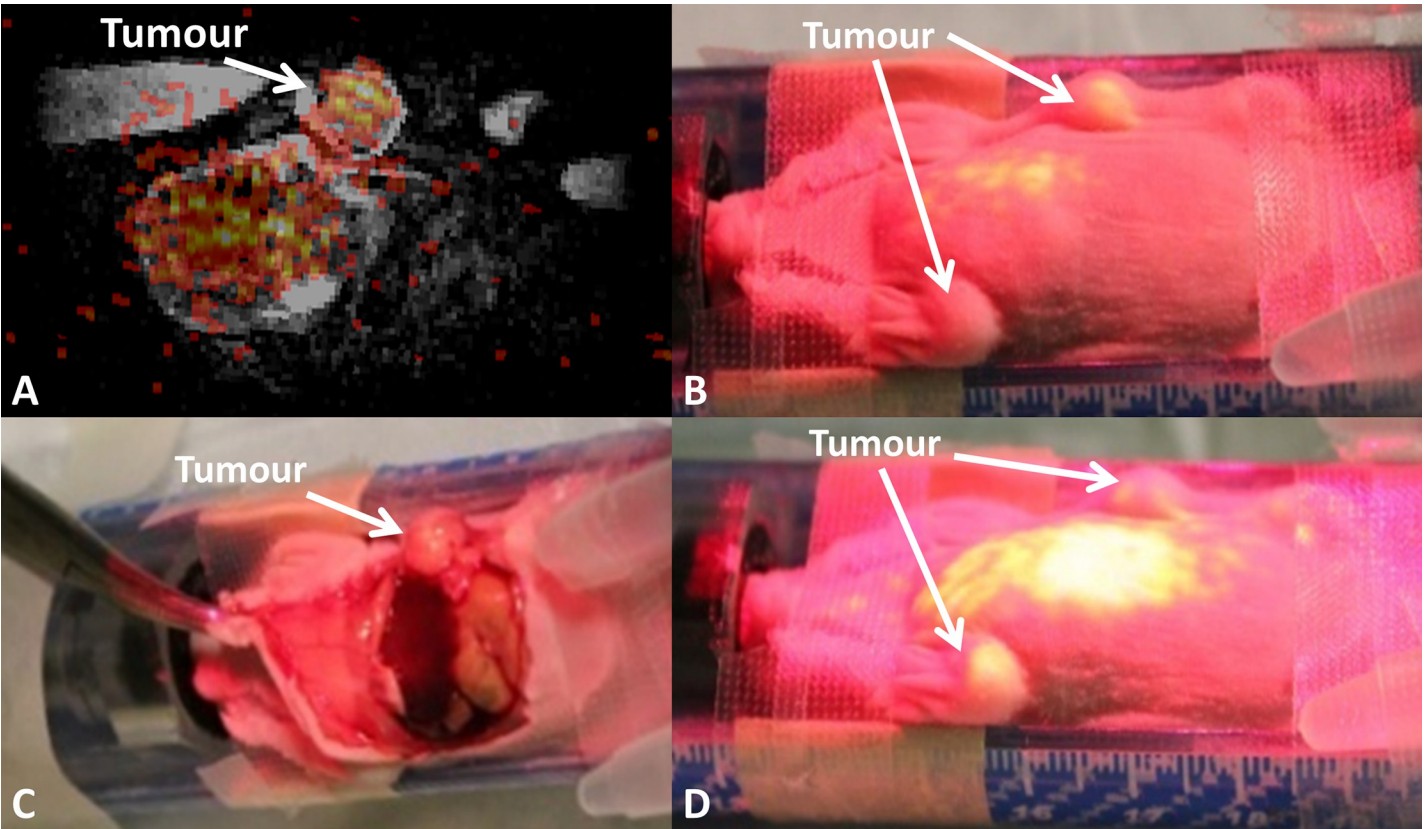

**Fig 4. Augmented reality guided procedure with the breast carcinoma xenograft model demonstrating breast tumors in the left and right mammary fat pads.** (A) A registered PET/MRI image coronal slice for AR guide procedure. (B) The AR PET image projected on the mouse's body surface highlights the tumor location. (C) The dissected mouse reveals the tumor location on the right in the same location. (D) AR projection on the mouse demonstrates the tumor location on the left side and the 64Cu isotope uptake in the lungs and liver.

where most of the existing projection based AR device operates separately from the navigation system and incapable to track surgical instruments [24–26]. Firstly, our AR system is fully migrated to surgical navigation system. Secondly, our AR system use of surgical tracking device which provide real-time localization therefore displacement of AR device does not require re-registration. S1 Video depicts the core technology of AR-SN system through artificial chest wall phantom with delineated tumour, the AR device is projecting virtual tumour and CBCT image slice according to tracked pointer location. Pointer is gradually moved to expose tumour location underneath the skin surface.

The rationale behind this novel system is that the projection of virtual images on the surgical field can provide a useful visual guide to localize the extent of tumour intraoperatively and to alert surgeons to the presence of critical structures including vasculature or nerves that may not be readily visible. The principle of the surgical safety alert is discussed in our published manuscript regarding skull base surgery [6, 16]. In developing this AR-SN system we hope to provide a tool to aid in challenging surgical ablations at high risk of incomplete resection and major complications due to complex anatomy.

Several AR devices have previously been reported in the surgical literature. The majority of these devices have been designed as HMDs, as first described in 1969 by Land and Sutherland in collaboration with the Department of Defence [27]. In 1994, advances in static and dynamic registration of an optical see-through HMD were reported [28]. Since then, optical see-through AR has been successfully implemented in a number of non-medical applications as

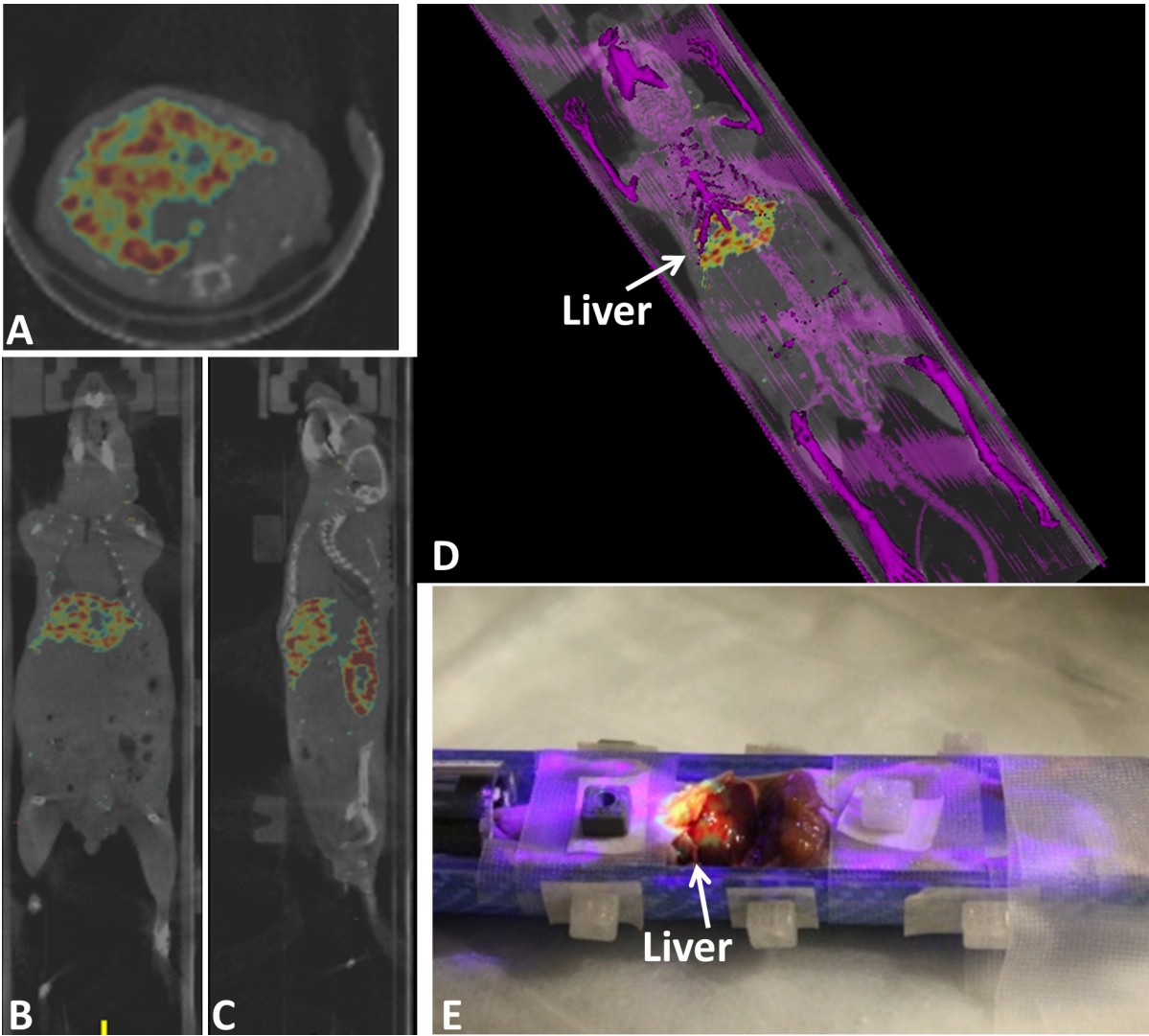

**Fig 5. Augmented reality guided procedure with the tumor bearing athymic CD-1 nude mouse using registered SPECT/CT image.** (A), (B) and (C) axial, coronal and sagittal images of fused SPECT/CT images respectively. SPECT signal is seen in rainbow and CT in gray. (D) an augmented reality projection image showing surface rendering of bone in purple combined with SPECT/CT coronal slice. (E) mouse dissection mouse demonstrating augmented reality projection SPECT-CT images overlaid on the mouse body also highlighting isotope uptake by the liver.

well as more recent medical applications in the field of neurosurgery and head and neck surgery [29–31]. However, HMDs have several drawbacks including the indirect view of the surgical field and extra equipment that crosses the surgeon's sightline and/or restricts head movement due to device wiring. The add-on weight of HMDs on the surgeon head could potentially reduce strength and concentration with prolonging usage of HMDs during the course of surgery.

To address these limitations, scientists proposed using image overlay systems, which provide an alternative approach to enhance surgical visualization. Weiss et al. developed an inexpensive image overlay system composed of a LCD monitor and a half-silvered mirror mounted on a MRI scanner, which could be adapted to various angles and procedures [32]. With this method, the surgical field is augmented by superimposing the images from the translucent mirror on to the operative field. However, with this system the operator is required to

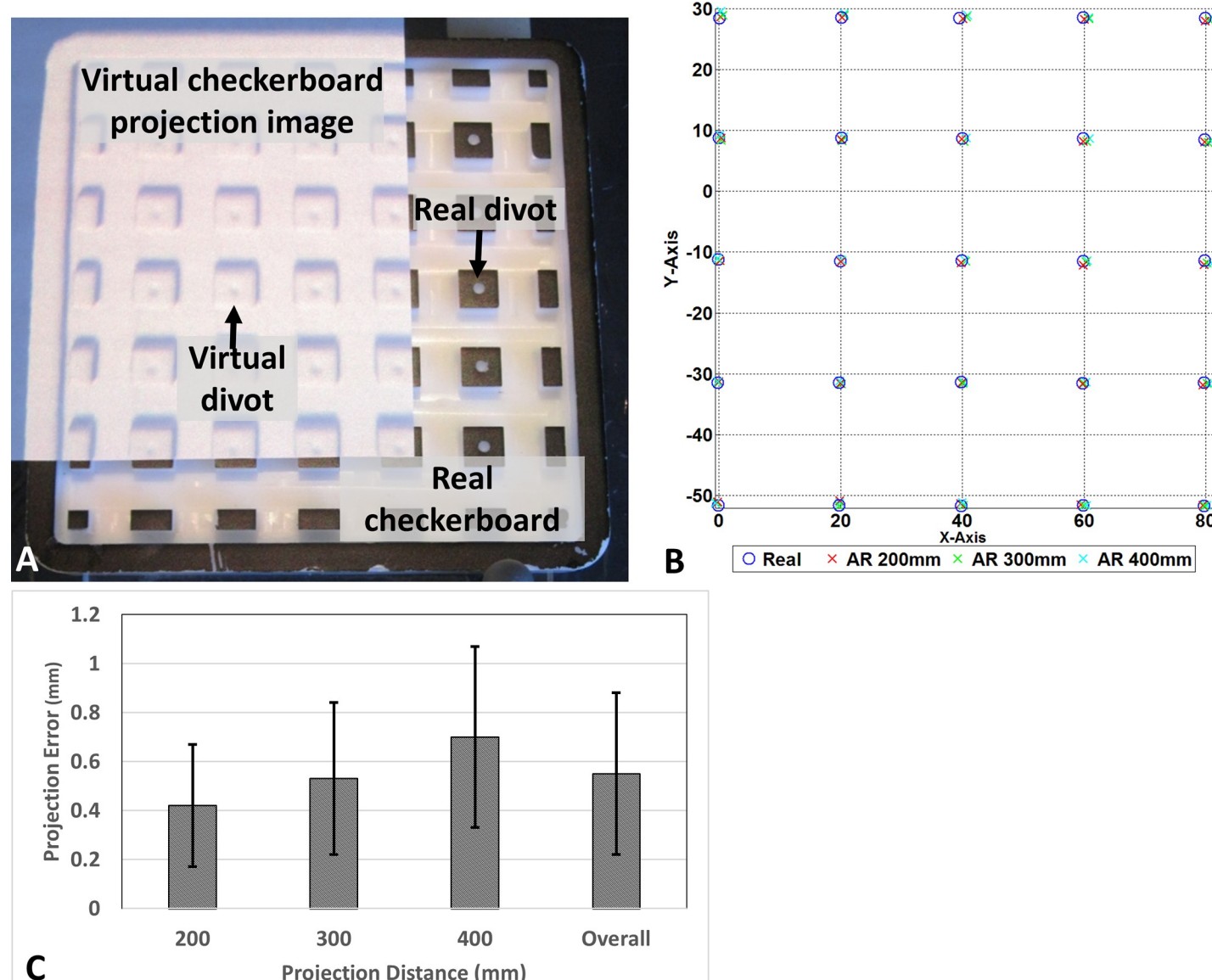

**Fig 6. Accuracy estimation of augmented reality navigation system.** (A) CBCT generated virtual checkerboard overlaid on a real checkerboard, (B) graphical representation of localization for each divot from various projection distances, (C) projection error at varies projection distance.

be stationary and the procedure must by accomplished close to the MRI scanner which may not be clinically practical. Baum et al. recently improved the versatility of this system by miniaturizing the monitor with a tablet device capable of operating independently from the scanner [33].

The next logical step forward for surgical AR is directly projecting the desired visual information onto the operative field. However, few papers have reported the application of this concept in the field of head and neck surgery [31, 34, 35]. Our preliminary results demonstrate that our AR-SN system can reliably and accurately augment visualization intraoperatively on both phantoms and animal models, while overcoming some disadvantages of existing systems. Our system was created using light equipment that can be easily adapted to the operative setting without any interference with the surgeons' sightline and working space. The capability of

visually enhancing the position of both the tumour and critical structures on the anatomical planes could allow for both accurate tumour delineation and the prevention of major complications. Moreover, our preclinical analysis demonstrated how the prototype could be adapted to several potential imaging sources (i.e. PET/CT, PET/MRI, SPECT/CT). This aspect is critically important as it provides the surgeon with a visual image based on radiologic and nuclear medicine semiotics (i.e. contrast agent uptake, inflammation- vs neoplastic-induced changes) and enables seamless incorporation of patient-specific medical information into an image-guided procedure. Meanwhile, the multi-image modality guidance can be very helpful in surgical navigation system with AR capability.

The checkerboard phantom study provided a quantitative proof of system accuracy, with over 225 points being localized with minimal error at various projection distances (0.55 ± 0.33 mm). This sub-millimeter accuracy is consistent with navigation system requirements and comparable to reported literature [36]. Our prototype system is based on a low cost video camera that serves as a sensor for the pico-projector calibration. This calibrated camera could be additionally used as a video see-through AR device to stream surgical video to the navigation platform. This potential to combine video into the pico-projector system could further enhance the functionality of our platform as a dual AR device.

There are a wide range of potential clinical applications for this type of AR-SN system. This technology may be beneficial for a broad spectrum of surgical procedures requiring sophisticated surgical planning, precise resection, and sparing of critical structures (e.g. spine surgery, chest wall surgery, orthopedic oncological surgery, and head and neck surgery). In the field of head and neck oncology, this technology could be applied to guide complex resections especially in areas where the bony framework substantially limits the motion of soft tissues during the surgical procedure. Furthermore, this technology could help identify small volume mass such as intraparenchymal lesions. The added clinical value of this AR-SN system is currently under investigation at the hybrid preclinical/clinical Guided Therapeutics Operating Room (GTxOR–TECHNA Institute, University Health Network, Toronto, Ontario, Canada) at our institution.

Our present study does have some limitations that we hope to overcome during future technology development. Firstly, our data is based on a preclinical study with a limited number of animals. Consequently, the highly controlled operative environment of animal models may not accurately reflect the clinical setting. Meanwhile, several literatures [37] had already demonstrated the value of AR technology in clinical setting. Secondly, the surgical site and anatomical structures in the pre-clinical studies are relatively motionless. This led us to assume that this system would be applicable only to areas where soft tissue morphological changes are limited by the bony framework (i.e. maxillofacial skeleton and surrounding spaces). In fact, our system does not currently have the capacity to account for tissue deformation induced by cauterization, manipulation, and resection. Therefore, procedures involving mainly soft tissues areas that are prone to significant deformation (i.e. lung) may not benefit from this AR-SN system. Finally, image distortion due to projection on non-planar surfaces is a further limitation of projection-based AR techniques. Despite these potential limitations, we are currently in the process of translating this technology to patient studies in key surgical applications such as head and neck and orthopedic oncology to evaluate system performance under clinical conditions.

## Conclusions

We have reported the development of a novel, integrated AR-SN system. This proof-of-concept study demonstrated the feasibility of our AR-SN system for multi-modality image-guided

surgery in a preclinical setting. The accuracy demonstrated from the phantom study was within acceptable uncertainty. Our AR-system was found to be highly precise and capable of sub-millimeter accuracy, which is in keeping with existing commercially available SN systems. These preliminary results represent a promising framework for future technology development and eventual clinical translation.

## Supporting information

**S1 Fig.**
(TIF)

**S1 Video.**
(MP4)

**S1 File.**
(DOCX)

## Acknowledgments

Thanks for Deborah A. Scollard, Kristin McLarty and Raymond M. Reilly's Lab for their assistant in microSPECT/CT study.

## Author Contributions

**Conceptualization:** Harley H. L. Chan.

**Data curation:** Harley H. L. Chan, Stephan K. Haerle, Jinzi Zheng.

**Formal analysis:** Harley H. L. Chan.

**Funding acquisition:** Jonathan C. Irish.

**Investigation:** Harley H. L. Chan, Stephan K. Haerle.

**Methodology:** Harley H. L. Chan.

**Project administration:** Harley H. L. Chan.

**Resources:** Harley H. L. Chan, Jinzi Zheng, Lauren Philp.

**Software:** Harley H. L. Chan, Michael J. Daly.

**Supervision:** Harley H. L. Chan, Jonathan C. Irish.

**Validation:** Harley H. L. Chan.

**Visualization:** Harley H. L. Chan.

**Writing – original draft:** Harley H. L. Chan.

**Writing – review & editing:** Harley H. L. Chan, Stephan K. Haerle, Lauren Philp, Marco Ferrari, Catriona M. Douglas.

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
