## [Decision Letter · Decision Letter 0]

2 Mar 2021

PONE-D-20-36797

An integrated augmented reality surgical navigation platform using multi-modality imaging for guidance

PLOS ONE

Dear Dr. Chan,

Thank you for submitting your manuscript to PLOS ONE. After careful consideration, we feel that it has merit but does not fully meet PLOS ONE’s publication criteria as it currently stands. Therefore, we invite you to submit a revised version of the manuscript that addresses the points raised during the review process.

Please consider the minor revisions required by Reviewer 1 and with those responses in view, I very much look forward to receiving the revision again.

We look forward to receiving your revised manuscript.

Kind regards,

Domokos Máthé

Academic Editor

PLOS ONE

Journal Requirements:

"This work is funded by the Guided Therapeutics (GTx) Program-TECHNA Institute, University Health

347 Network, Kevin and Sandra Sullivan Chair in Surgical Oncology, Hatch Engineering Fellowship Fund, and

348 Princess Margaret Hospital Foundation. IDEAS grant by Radiation Medicine Program, Princess Margaret

349 Cancer Centre. Thanks for Deborah A. Scollard, Kristin McLarty and Raymond M. Reilly for their assistant

350 in microSPECT/CT study."

Additional Editor Comments:

Thank you for bearing with us in your continued patience as so many reviewers could not cope with the multi-disciplinary nature of your manuscript. I myself, having been able to participate in previous similar medical device developments also very much appreciate the manuscript and its results. Please kindly refer to the requests of the one reviewer requiring minor, albeit important improvements, too.

Reviewers' comments:

Reviewer's Responses to Questions

**Comments to the Author**

1. Is the manuscript technically sound, and do the data support the conclusions?

Reviewer #1: Yes

Reviewer #2: Yes

2. Has the statistical analysis been performed appropriately and rigorously? 

Reviewer #1: I Don't Know

Reviewer #2: Yes

3. Have the authors made all data underlying the findings in their manuscript fully available?

Reviewer #1: Yes

Reviewer #2: Yes

4. Is the manuscript presented in an intelligible fashion and written in standard English?

Reviewer #1: Yes

Reviewer #2: Yes

5. Review Comments to the Author

Reviewer #1: As I know this is the firts proof-of-concept study which demonstrated the feasibility of a system for multi-modality image-guided surgery in a preclinical setting. This methods may be a very important testing platform of the development of image-guided surgery especially the spatial precision which will be very helpful in par example neurosurgery.

I have some questions:

1) Fig 3. shows the result of projection of the PET/SPEC/CT image of the tumor on the skin and tomour of the mouse. But the image (see Fig 6 B-D) is obviously mistaken because the image or tumour is projected to the bed. How could you develop the projector to reduce this mistake?

2) Based on the description, the spatial resolution of the system is not clear. But Fig 6. try to visualise the arrangement but the mistake of measurement along the x-y-z axis is not shown. How could you determine these parameters?

3) How could you develope a rendering methods to reduce the real resolution problem of the molecular imaging modality? How could joint to the PET/CT images also to reduce this type problems?

4) I suggest to specify the limitation of this proof-of-concept very thoroughly and please comlete how you will solve these.

Reviewer #2: Authors demonstrated that the feasibility of their novel augmented reality system with pico-projector which was fully integrated with a real-time surgical navigation system. They also showed how their prototype could be adapted to several different imaging sources (PET/CT, SPECT/CT, PET/MRI). Their phantom study provided a quantitative proof of system accuracy with minimal error at various projection distances which might be acceptable in real clinical setting.

As they mentioned, there are some limitations. I think that major drawback would be image distortion on non-planar surface and morphological tissue change during operation. So this technology could be applied to relatively hard and non-deformable organ such as brain and bony framework.

Even if there are some limitations, this article would be readable for surgeons or physicians who are interested in image guided surgery.

6. PLOS authors have the option to publish the peer review history of their article (what does this mean?). If published, this will include your full peer review and any attached files.

Reviewer #1: No

Reviewer #2: No

---

## [Author Response · Author response to Decision Letter 0]

31 Mar 2021

PONE-D-20-36797

An integrated augmented reality surgical navigation platform using multi-modality imaging for guidance

Response to reviewers:

Reviewer #1: As I know this is the first proof-of-concept study which demonstrated the feasibility of a system for multi-modality image-guided surgery in a preclinical setting. This methods may be a very important testing platform of the development of image-guided surgery especially the spatial precision which will be very helpful in par example neurosurgery.

I have some questions:

1) Fig 3. shows the result of projection of the PET/SPEC/CT image of the tumor on the skin and tumor of the mouse. But the image (see Fig 6 B-D) is obviously mistaken because the image or tumour is projected to the bed. How could you develop the projector to reduce this mistake?

 Thank you for your comment. We agree with the reviewers regarding the misleading information of the virtual tumor being projected onto the mouse’s bedding. Indeed, a portion of the tumor is behind the imaging bed with respect to the projection direction. The AR-SN system follows the instruction correctly projecting the virtual tumor below the projection surface. However, the obstruction in the space between the projector’s prospective to the subject’s surface may not be corrected by the AR-SN system. The attached figure (axial slice from same pre-clinical animal study) demonstrates the cause of the misleading projection on the imaging bed and fiducial marker. Additionally, the semi-translucent 3D printed fiducial marker is diffusing projection light to the surrounding area that further extends the projection error. We have recently started a comprehensive upgrade of our AR-SN system. One of the enhanced abilities will be projecting registration landmarks which will hopefully help to reduce errors in projection. The new abilities of our system allows for fine adjustments of the AR image including scale factor, horizontal and vertical shift. Prior to using the AR-SN system, the user will be able to project registration landmarks and check their position with respect to fiducial markers and makes any image adjustment if require. This fine adjustment function will able to mitigate systematic error. Regarding the surface topology-induced projection error, if we consider the size of tumors in humans as well as the clinical scenarios in which AR-SN would be used (i.e., projection of a tumor onto a rather flat surgical field to provide surgeons with an on-the-field representation of tumor extension) the image distortion will likely be very minimal.

2) Based on the description, the spatial resolution of the system is not clear. But Fig 6. try to visualize the arrangement but the mistake of measurement along the x-y-z axis is not shown. How could you determine these parameters?

 Thank you for your question. We agree with the reviewers comments on error measurement. The sentence below is amended in the manuscript (lines 176 to 180) to explain further how to compute root-mean-square (RMS) from the system (x,y,z) uncertainty. 

“The uncertainty of the projected location was calculated as follows: (xi, yi, zi)real represents as “real” location of the divot center acquired manually without AR image guidance and (xi, yi, zi)virtual represents as “virtual” location of the divot generated from the AR projection image, the estimation of error is the distance between the “real” and “virtual” location of the central divot such that Err=‖(x_i,y_i,z_i )_real-(x_i,y_i,z_i )_virtual ‖.”

3) How could you develop a rendering methods to reduce the real resolution problem of the molecular imaging modality? How could joint to the PET/CT images also to reduce this type problems?

 Thank you for your interesting question. The PET/CT image is fused by rigid registration using normalized mutual information. The opacity of the fused image can be adjusted in order to highlight either the PET signal or CT image. To reduce real resolution problems, the voxel value is interpolated in order to mitigate pixilation due to low the resolution of the PET image. Additionally, a Gaussian smoothing filter can be applied to the surface rendering of virtual objects to improve resolution. However, these proposed solutions will only able to minimize and not completely resolve the issue of low image resolution. 

4) I suggest to specify the limitation of this proof-of-concept very thoroughly and please complete how you will solve these.

 Thank you for reviewer’s suggestion. In the manuscript from line 317 to 330 the limitations of our AR-SN system have been specified. The major challenge in the development of the AR-SN system is projecting the correct image onto an irregular three-dimensional surface as the image can become distorted. In machine vision, methods to correct this image distortion have been widely investigated. Camera calibration of intrinsic / extrinsic parameters is one of the common approaches to which can used to rectify distorted images. A similar approach can be applied to try to correct the projected image. However, the fundamental problem of image distortion originates from the surface topology of the subject. In order to correct this distortion, the surface topology needs to be characterized. This can be achieved by segmenting the surface from the CT image then interpolating the surface by best fitting B-spline or non-uniform rational B-splines (NURES). Once the topology of the surface is formulated, the distortion can be corrected by mathematically mapping the individual pixels to the formulated surface. This described approach is computationally intensive. Our AR-SN system has a refresh rate of 30 fps and thus the system may experience lag time issues with this approach. To minimize the lag problem, graphics processing unit (GPU) implementation can be considered.

Reviewer #2: Authors demonstrated that the feasibility of their novel augmented reality system with pico-projector which was fully integrated with a real-time surgical navigation system. They also showed how their prototype could be adapted to several different imaging sources (PET/CT, SPECT/CT, PET/MRI). Their phantom study provided a quantitative proof of system accuracy with minimal error at various projection distances which might be acceptable in real clinical setting.

As they mentioned, there are some limitations. I think that major drawback would be image distortion on non-planar surface and morphological tissue change during operation. So this technology could be applied to relatively hard and non-deformable organ such as brain and bony framework. Even if there are some limitations, this article would be readable for surgeons or physicians who are interested in image guided surgery.

 Thank you for your valuable comments and feedbacks. We realize that the limitation of our prototype system is that it may underperform when the image is projected onto deformable soft tissue. We are currently investigating potential solutions to this problem to improve the utility of our system.

---

## [Editor Report · Decision Letter 1]

12 Apr 2021

An integrated augmented reality surgical navigation platform using multi-modality imaging for guidance

PONE-D-20-36797R1

Dear Dr. Chan,

We’re pleased to inform you that your manuscript has been judged scientifically suitable for publication and will be formally accepted for publication once it meets all outstanding technical requirements.

Kind regards,

Domokos Máthé

Academic Editor

PLOS ONE
---

## [Editor Report · Acceptance letter]

22 Apr 2021

PONE-D-20-36797R1 

An integrated augmented reality surgical navigation platform using multi-modality imaging for guidance 

Dear Dr. Chan:

I'm pleased to inform you that your manuscript has been deemed suitable for publication in PLOS ONE. Congratulations! Your manuscript is now with our production department. 

Kind regards, 

on behalf of

Dr. Domokos Máthé 

Academic Editor

PLOS ONE